# A DNA Vaccine Encoding *Plasmodium falciparum* PfRH5 in Cationic Liposomes for Dermal Tattooing Immunization

**DOI:** 10.3390/vaccines8040619

**Published:** 2020-10-20

**Authors:** Wesley Luzetti Fotoran, Nicole Kleiber, Christiane Glitz, Gerhard Wunderlich

**Affiliations:** 1Departamento de Parasitologia, Instituto de Ciências Biomédicas II, Universidade de São Paulo, São Paulo 05508-000, Brazil; wesleylfw@hotmail.com (W.L.F.); nicole.kleiber@outlook.com (N.K.); 2Department of Molecular Physiology, Institute of Animal Physiology, Westfälische Wilhelms University of Münster, 48149 Münster, Germany; christiane.glitz@gmail.com

**Keywords:** nucleic acid vaccines, cationic liposomes, in vivo delivery, intradermal immunization, tattooing

## Abstract

Vaccines are the primary means of controlling and preventing pandemics and outbreaks of pathogens such as bacteria, viruses, and parasites. However, a major drawback of naked DNA-based vaccines is their low immunogenicity and the amount of plasmid DNA necessary to elicit a response. Nano-sized liposomes can overcome this limitation, enhancing both nucleic acid stability and targeting to cells after administration. We tested two different DNA vaccines in cationic liposomes to improve the immunogenic properties. For this, we cloned the coding sequences of the *Plasmodium falciparum* reticulocyte binding protein homologue 5 (PfRH5) either alone or fused with small the small hepatitis virus (HBV) envelope antigen (HBsAg) encoding sequences, potentially resulting in HBsAg particles displaying PfRH5 on their outside. Instead of invasive intraperitoneal or intramuscular immunization, we employed intradermal immunization by tattooing nano-encapsulated DNA. Mice were immunized with 10 μg encapsulated DNA encoding PfRH5 alone or in fusion with HBsAg and this elicited antibodies against schizont extracts (titer of 10^4^). Importantly, only IgG from animals immunized with PfRH5-HBs demonstrated sustained IgG-mediated inhibition in in vitro growth assays showing 58% and 39% blocking activity after 24 and 48 h, respectively. Intradermal tattoo-vaccination of encapsulated PfRH5-HBsAg coding plasmid DNA is effective and superior compared with an unfused PfRH5-DNA vaccine, suggesting that the HBsAg fusion may be advantageous with other vaccine antigens.

## 1. Introduction

Liposomes are artificial lipid-based nanoparticles with a round shape consisting of one or more phospholipid bilayer membranes that encapsulate an internal aqueous compartment [1]. They are easy to prepare and are formulated using non-toxic phospholipids and cholesterol, meaning that they are biodegradable, biocompatible and present low toxicity [2]. Liposomes can be classified according to their size and number of lipid bilayers. Small unilamellar vesicles (SUVs) are typically under 50 nm, while large unilamellar vesicles (LUVs) range between 50 and 500 nm, and giant unilamellar vesicles (GUVs) are capable of reaching up to 100 µm [3]. Particle size is especially important when designing synthetic vaccines, since size affects dispersion and diffusion speed, draining to the lymph nodes, and antigen concentration [4], which finally influences successful intracellular uptake by antigen-presenting cells. Small particles (<50 nm) are able to diffuse rapidly and can carry up to 10 antigen molecules. These are especially suitable for the delivery of nucleic acids, especially when using cationic liposomes [5].

The development of nucleic acid vaccines began not only as an attempt to avoid vaccines with live or attenuated vectors, but also as an investigation of the effects provoked by the direct injection of DNA expression vectors [6]. Nucleic acid-based vaccines do not rely on producing and purifying recombinant antigens, making them a simpler option with the potential for swift adaptation for any desired antigen, when compared to the production of traditional vaccines. Plasmid DNA-based vaccines are still the most widely studied form of nucleic acid vaccines and immunization with purified circular plasmid DNA has already been shown to be successful in small animals in various tissues and different routes of administration [6,7,8,9]. However, the immunogenicity and effectiveness are still only moderate in human applications (for example see [10]).

Skin tattooing is a novel approach in intradermal immunization and has already been applied to DNA vaccination. DNA skin tattooing induced an even stronger immune response than intramuscular needle injection supported by adjuvants [11]. Some key advantages make skin tattooing an interesting choice for intradermal immunization: (i) it covers a large skin area, which could potentially elicit a stronger immune response, especially given that it is a region abundant in antigen-presenting cells; (ii) it is inexpensive; (iii) it produces low DNA damage, and (iv) it recruits immune cells, like macrophages, which enhances the immune priming [12]. Here, we apply dermal tattoo-immunization using small amounts of a DNA vaccine associated with cationic liposomes to enhance plasmid delivery. We included the *Plasmodium falciparum* reticulocyte binding protein homologue 5 (PfRH5), an antigen of *P. falciparum* merozoites, which is essential for the invasion of human red blood cells by recognition and interaction with Basigin [13]. Antisera raised against PfRH5 were tested in in vitro assays to analyze the reinvasion blocking potential of antibodies. Additionally, we compared the effect of immunizing plasmids encoding the antigen PfRH5 only and also in fusion with the small hepatitis B virus envelope antigen (HBs) coding sequence, which in theory leads to a virus-like particle when produced in transfected cells. Our data suggest that there was an efficient humoral response when PfRH5 was delivered in fusion with HBs.

## 2. Materials and Methods

### 2.1. Cloning of Plasmid Constructs

The PfRH5/HBs expressing plasmids used were previously described [14]. pcDNA3-GFP, which drives green fluorescent protein expression, and pcDNA3-Luc, encoding *Photinus* luciferase, were used as controls [14]. All plasmids were purified from *E. coli* DH10B cells using standard molecular techniques and produced in large scale using a plasmid preparation protocol for high purity [15].

### 2.2. Liposome Preparation and Entrapment of Plasmid DNA

All chemicals were from Sigma-Aldrich/Merck or Carl Roth (Darmstadt, Germany). All lipids were from Avanti Polar Lipids (Alabaster, Alabama, USA). Liposomes were prepared using DDAB (Dimethyldioctadecylammonium bromide) and DC-cholesterol (3β-[N-(N’,N’-dimethylaminoethane)-carbamoyl]cholesterol hydrochloride) in a molar ratio of 4:1. Then, both DDAB, cholesterol and DSPE-PEG2000 (1,2-Dimyristoyl-*sn*-Glycero-3-Phospoethanolamine-N-[Methoxy(Polyethylene glycol)-2000, ammonium salt) were dissolved in 1 mL chloroform and left under a constant N_2_ flow for the elimination of chloroform and formation of a phospholipid film on the tube walls. This film was then further dried under vacuum for at least 1 h. The chloroform-free film was then rehydrated in 1 mL 5 mM Tris-HCl, pH7.5 at 60 °C (DDAB transition temperature) for one hour and stirred vigorously at 10-minute intervals. The resulting opaque solution was subsequently subjected to sonication at high energy until it was transparent, followed by centrifugation for one hour at 100,000 g. The supernatant containing unilamellar lipid vesicles was used for the preparation of the liposomal formulation.

The cationic liposomal nanoparticles were loaded with target DNA using different amounts of plasmid per nmol of lipid (8 nM DDAB lipids per µg plasmid, 1 µg of pcDNA3-PfRH5-HBs equals ~0.2 pmol) by seven alternating cycles of freezing in liquid nitrogen for 1 min and thawing in a water bath of 60 °C for 1 min. Afterwards, the formulation was passed repeatedly (at least seven times) through polystyrene membranes with pore sizes of 0.2 to 0.8 μm (extrusion).

### 2.3. Cryoelectron Microscopy

Liposomes were prepared as described and analyzed by cryo-electron microscopy as previously described in [14]. Briefly, 1 mL plasmid DNA-loaded liposomes were applied over a holey carbon-film grid (QuantifoilMicro Tools, Jena, Germany), followed by flash-freezing in liquid ethane using a Gatan Cryoplunge 3 (Gatan, Pleasanton, CA, USA). The visualization of the frozen, hydrated specimen was performed on a JEM2100 electron microscope (JEOL, Tokyo, Japan, operating at 200 kV) resulting in images of 2 μm scale. Image recording was done with a Gatan Ultrascan 4000 CCD camera at 40,000× magnification.

### 2.4. Liposome Transfection In Vitro and In Vivo Assay

HEK293T and 786-O cells (ATCC, Manassas, VA, USA) were seeded overnight in a 24-well plate at a density of 1 × 10^4^ cells/well and transfected with liposomes (8 nM lipids/µg DNA) with 1 µg of total DNA/well. After 24 h incubation, the cells were washed with DMEM medium, trypsinized and centrifuged for 5 min at 1500 rpm. The pellets were resuspended in PBS and analyzed for GFP expression by flow cytometry (Guava easycyte). In vivo assays were done by intraperitoneal inoculation of 10 µg pcDNA3-GFP (8 nM lipids per µg DNA). After 24 h, the peritoneal region from euthanized mice was washed with PBS and the cells were recovered and analyzed by flow cytometry. The samples checked for luciferase expression were treated with Brefeldin A (BD GolgiPlug^™^), according to manufacturer’s instructions, 6–8 h before every analysis. The cells were pelleted at 14,000 rpm for 20 s and resuspended in Luciferase Assay Substrate and Buffer, according to the manufacturer’s instructions (Promega), before analysis in a Berthold Lumat luminometer.

### 2.5. Intradermal Immunization

All experiments involving mice were conducted after approval of the experimental protocol (Protocol number 76/2016, Commission for Ethics in Animal experimentation at ICB/USP). Five to twelve week-old BALB/C mice were obtained from the breeding facility for isogenic mice at the Department of Parasitology (ICB/USP) and kept under pathogen-free conditions during the course of the experiment. Groups of BALB/C mice were sedated with ketamine/xylazine and had their hind limb hair removed with commercial hair removal cream and the skin was sterilized with 70% ethanol. For the delivery of plasmid DNA, 30 µL of a liposome/pDNA mixture at 8 nM lipid concentration were administered in two drops on the hairless skin on the tibia anterior muscle followed by tattooing of a 2 × 1 cm skin area using a commercial tattoo machine. The tattoo device was adjusted to expose no more than 2 mm of the needle and used twice in the muscle for no more than 15 s at a voltage of 16 V set on the power supply. Thus, every mouse received 10 µg of DNA in 50 µL liposome solution in the course of one immunization. As some trauma was caused to the skin, a silicone cream was applied to the tattooed area. Using 10 μg of DNA in 50 μL for each immunization, five animals per formulation were immunized on the skin on days 0, 14, and 28. Blood samples were recovered on day 0 (pre-immune sera), 14, 28 and 42. To determine antibody titers, the sera from all four blood harvests were tested for reactivity against *P. falciparum* schizont extracts by standard ELISA [16]. Sera were also used for Western blot and IgG purification for posterior growth inhibition assays. For in vivo imaging, treated animals were bio-imaged in an IVIS^®^ Spectrum CT (Caliper Life Sciences), located at the CEFAP-ICB/USP. The tattooed area was analyzed for bioluminescence (for luciferase), according to the IVIS manual.

### 2.6. P. falciparum Culture and Invasion Inhibition Assays

*P. falciparum* strain NF54 was maintained in human B+ red blood cells (hematocrit 5%) in RPMI medium supplemented with 10% human B plasma or 0.5% Albumax 1 (Invitrogen/ThermoFisher Scientific, Carlsbad, CA). Permission for the use of human blood and plasma for these experiments was obtained from the local committee for ethics in research involving humans or human samples at ICB-USP (CEPSH-ICB/USP, protocol No. 874/2017). Cultures were kept in candle jars in an incubator at 37 °C with daily medium change [17]. Parasitemia was monitored by Giemsa-stained thin blood smears (Panótico Quick kit, LaborClin, Pinhais, Brazil) and microscopy at 1000 × magnification. Blood stage parasites were synchronized by intermittent plasmagel flotation [18] (Voluven 6%, Fresenius-Kabi, Campinas, Brazil) followed by sorbitol lysis [19], and then plated in 96 well plates at 1% initial parasitemia. For growth inhibition assays (GIA), protein A-purified IgG fractions from pre-immune and immunized mice were added at different concentrations (0.180 μg/mL and 300 μg/mL) and volumes were adjusted with RPMI medium. Parasitemia were monitored after 24 h and 48 h. To measure the parasitemia, 5 µL culture aliquots were removed from wells and stained with ethidium bromide, followed by flow cytometry (Guava easycyte, Merck-Millipore, Darmstadt, Germany) as described before in [20]. Additionally, standard blood smears were done for confirmation. The percentage of inhibition in immune IgG-treated parasites was calculated using the triplicate parasitemia readouts in relation to the pre-immune IgG-treated cultures, applying the following formula:% Inhibition = (1 − parasitemia (immune IgG treated cultures)/parasitemia (pre-immune IgG treated cultures)) × 100

### 2.7. ELISA and Western Blots

For the detection of PfRH5-reactive IgG, ELISA plates (medium binding, Jet Biofil, Guangzhou, China) were coated with 200 ng/well late schizont stage extracts at 4 °C overnight in 50 mM carbonate buffer (pH 9.6). Then, plates were rinsed with PBS/Tween 0.05% (PBS/T) and blocked with 2% skimmed milk/PBS for 1 h at room temperature. Wells were then washed twice with PBS/T followed by a 2 h incubation at room temperature with antisera generated in mice previously immunized with encapsulated pcDNA3-PfRH5-HBs or pcDNA3-PfRH5. As controls, pre immune sera from mice of each group were used. For titer determination, antisera were endpoint-diluted. After four washing steps with PBS/T, a peroxidase-coupled anti-murine-IgG diluted 1:2000 (KPL-Seracare, Milford, MA, USA) was applied for one hour at room temperature. After thorough rinsing (4 washes), wells were developed with TMB substrate (Pierce/ThermoFisher Scientific, Sao Paulo, Brazil) and the colorimetric reaction was stopped after 30 min with 1 M HCl. The result was analyzed in a BioTek plate reader (BioTek, Winoosky, VT, USA) at 450 nm/595 nm.

For Western blots, 5 µg of schizont extract or recombinant HBsAg (a kind gift from the HBsAg production plant at the Butantan Institute, São Paulo, Brazil) were separated under non-reducing conditions in 10% SDS-polyacrylamide gels. Then, proteins were transferred on to nitrocellulose membranes (Hybond C, GE Healthcare, São Paulo, Brazil) and these were blocked with 2% milk in PBS/T for 1 h at room temperature. After two washing steps with excess PBS/T, a serum pool of encapsulated plasmid-immunized animals was incubated for one hour at room temperature at a 1:500 dilution in 1% milk PBS/T. After five washes with PBS/T, membranes were incubated with peroxydase-conjugated anti-mouse IgG secondary antibody diluted to 1:2000 (KPL) for one hour at room temperature. After five washes with PBS/T, the membranes were briefly soaked in ECL reagent (GE Healthcare) and chemiluminescence was documented using Hypermax X-ray films (Kodak, Rochester, NY, USA) or photographed using a GE Image Quant 3000 apparatus (General Electric Healthcare, Chicago, IL, USA).

### 2.8. Statistical Analyses

Raw data were analyzed using Graph Pad Prism 5.03 (Graph Pad Software, San Diego, CA, USA) and Origin 8 (OriginLab Corporation). Student’s *t*-test was used to compare normally distributed values between groups and one-way ANOVA or a Kruskal–Wallis Test with post hoc correction was used to compare three or more groups (*p* < 0.05 was considered statistically significant). An analysis using non-linear regression as dose-response curves was performed to determine levels of liposome toxicity and to compare transfection efficiency between groups.

## 3. Results

### 3.1. Characteristics and Morphology of DDAB/DC-Chol Liposomes and Liposomes/DNA Complexes

First, we characterized the DDAB-Chol liposome formulations before (Figure 1A, liposomes alone) and after (Figure 1B, liposomes loaded with DNA) encapsulating 1 μg of DNA for each 8 nM of total lipids by measuring their zeta potential, polydispersity index and diameter using a dynamic light scattering analyzer (Figure 1A, Table 1). As expected, the zeta potential decreased after the addition of DNA, forming DNA/lipid complexes that increased the size of the original liposomes. The liposomes turn into a rigid form. The observed polydispersity indices (PDIs) were close to 0.2, which is consistent with a monodisperse particle size distribution. The diameter of the liposomes was estimated to be smaller than 100 nm but were in fact slightly larger, with a diameter of around 100 nm, as seen in Cryo-TEM (Figure 1). After plasmid-loading, the diameter of the liposomes increased as expected from previous works [14].

### 3.2. Transfection In Vitro and In Vivo Reveals Efficient Antigen Production in Target Cells

The transfection efficiency of DNA entrapped in liposomes was analyzed in two different cell lines. The plasmid pcDNA3-GFP encoding the GFP protein was used initially in three conditions: pcDNA3-GFP alone, pcDNA3-GFP in cationic liposomes and liposomes alone. To verify the GFP fluorescence, transfected HEK293T and 786-O cells were analyzed by flow cytometry. In addition, cells from BALB/C mice intraperitoneally immunized with cationic liposome-packaged pcDNA3-GFP were washed with PBS 24 h after immunization and measured accordingly. The gates of HEK293T 24 h after transfection are shown in Figure 2A. The fluorescence detected in cells transfected with liposomes alone and transfected with DNA in cationic liposomes are shown in Figure 2B and Figure 2C, respectively. While only background signals were found in cells transfected with empty liposomes, HEK293T cells 786-O cells as well as spleen cells from intraperitoneally immunized mice showed a strong GFP signal in around 15% of the measured cells (Figure 2D).

We then asked if the transfection is caused by DNA entrapped inside the cationic liposomes or by DNA attached via its charge to the outside of the cationic vesicles. The cationic liposomes were conjugated to pcDNA3-Luc and followed by DNAse treatment. The liposomes were incubated with HEK293T cells and 24 h later, cell extracts were analyzed for luciferase activity. Cationic liposomes treated with DNAse were still able to transfect by protecting the entrapped plasmid. Interestingly, DNAse treated DNA/liposomes increased the transfection efficiency in vitro, suggesting that plasmids associated on the outside of vesicles actually hinder the liposome’s access to cells (Figure 2E). In order to test if the less invasive intradermal immunization by tattooing was equally effective, animals were submitted to tattooing in the left leg with pcDNA3-GFP packaged into cationic liposomes. Under IVIS detection, the animals showed GFP expression on exposed skin in the previously treated area, demonstrating that the dermal delivery of the particles was also functional, transfecting cells and producing fluorescent GFP in the tattooed tissue (Figure 2F).

### 3.3. Intradermal Tattooing Immunization against PfRH5-HBs Generates Functional Antibodies against P. falciparum

In the next step, we tested the ability of the tattoo technique to elicit antibodies against the malaria vaccine candidate PfRH5. For this, five animals per group were tattooed with plasmid constructs encoding PfRH5 in a secreted form or with the PfRH5 fused to HBs, which is supposed to form a secreted virus-like particle after synthesis in cells. Sera from immunized animals were able to recognize PfRH5 in schizont extracts in Western blot and ELISA assays (Figure 3). As expected, only animals immunized with PfRH5 fused to HBs also reacted with HBs protein (data not shown). After three immunizations, antibody titers of around 10^4^ against schizont extracts were observed, while the pre-immune sera showed only base-line levels of antibodies in this test. Immunization with pcDNA3-PfRH5-HBs resulted in slightly higher (not significant) titers than immunization with pcDNA3-PfRH5. To address the question of whether these antibodies were also able to inhibit reinvasion of merozoite in red blood cells in a dose-response manner, IgGs from immunized animals were purified and added to in vitro *P. falciparum* cultures. As controls, parasites were grown in the presence of pre-immune IgG in the maximum quantity of 300 µg/mL. After 24 h and 48 h, the culture parasitemia was analyzed to calculate parasite growth inhibition versus the control cultures. IgGs from animals immunized only with PfRH5 could only block the reinvasion in vitro for up to 24 h of culture (average 31.4% +/− 4.9 for 180 µg/mL and 54.1% +/− 8.2 for 300 µg/mL). However, animals immunized with PfRH5-HBs showed sustained inhibition. This was observed even after 48 h, suggesting that PfRH5-HBs enhanced the vaccine effect in animals tattooed with DNA encoding this construct (Figure 3C, average 38% +/− 2.5 for 180 µg/mL and 54.1% +/− 4.2 for 300 µg/mL in 24 h and average 23.4% +/− 8 for 180 µg/mL and 39% +/− 13.4 for 300 µg/mL in 48 h). In contrast, IgG from animals immunized solely with pcDNA3-PfRH5 were no longer inhibitory after 48 h incubation (Figure 3C).

## 4. Discussion

The improvement of immunogenicity and speeding up vaccine production are issues of pivotal importance in many infectious diseases. For example, in the SARS-CoV-2 pandemic these are of utmost urgency in order to save lives and protect infected persons from possible sequelae [21]. In this regard, nucleic acid vaccines are promising candidates due to their potentially faster production and easier upscaling [22,23]. Nevertheless, DNA and RNA vaccines still seem to be less immunogenic than conventional protein-based vaccines. A critical problem is how to enhance transfection and entry into cells/nuclei of cells when applying DNA vaccines [24,25] or how to stabilize and protect RNA molecules until their delivery into the cytoplasm [26,27]. Here, we focused on the delivery of DNA loaded in cationic liposomes in order to improve transfection efficiency. Previous work had already shown the feasibility of packaging DNA into liposomes for vaccine applications, however, the very invasive peritoneal route was employed [14]. In order to test a more amenable way of immunization, we used the intradermal route for delivery and applied a tattooing device, previously suggested by other groups [12,28,29,30].

When first testing the transfection efficiency in cultured eucaryotic cells, we observed that previous DNAse treatment increased transfection efficiency, probably due to the negative charge of DNA on the outside of otherwise cationic liposomes. Although not tested for, it can be expected that DNAses contained in serum rapidly digest DNA not trapped inside liposomes, thus re-establishing their affinity for cell surfaces. Using the promising antigen PfRH5 [31], which is present in merozoites during blood stage malaria infection, we could create a dose-response inhibition with antibodies obtained from immunized animals [14,32]. Notably, the strategy for immunization is an important issue in inducing an enhanced immune effect. When animals were immunized with a plasmid encoding PfRH5, we obtained almost similar levels of antibodies when compared with animals immunized with plasmids encoding PfRH5 fused to HBs. However, the inhibitory effect of purified IgG clearly decreased after 24 h, which was less pronounced when PfRH5 was delivered fused to HBs. Possibly, the putatively higher density of PfRH5 on HBs-formed subviral particles compared with free PfRH5 and the consequent better display of the PfRH5 polypeptide, followed by the increased production of antiPfRH5 is responsible for this effect. Further, the quantity of truly inhibitory antiPfRH5 IgG may still be limited in the sera, explaining why a decrease in inhibition after 48 h incubation was observed. This may also explain the lack of any inhibitory effect in in vitro growth assays with sera from pcDNA3-PfRH5-immunized mice after 48 h incubation. Of note, Douglas and colleagues identified a number of monoclonal antibodies against PfRH5, which were not neutralizing or inhibitory [33]. In our experimental design, only 10 µg plasmid complexed to cationic liposomes was delivered in a total volume of 50 µL, and this formulation apparently had a greater capacity to transfect compared with DNA tattooed alone [28], as seen by the ROI, which increased by an order of two. We did not expect that in fact all plasmids applied to the skin were delivered to cells. However, the relatively inoffensive tattoo immunization was sufficient to elicit a robust humoral response against PfRH5. Taking into account that titers of antibodies were similar between studies, this route of delivery therefore appears similarly efficient as intraperitoneal immunization [14] with the difference that only three doses of tattoo-applied DNA-liposomes elicited the same antibody response as four i.p. immunizations. This underscores the ability of skin-resident cells to initiate a strong immune response [34]. In part, the problem of delivering plasmid encoding antigens can be overcome by the use of a microneedle and a polymer for the most efficient skin delivery [35]. Unlike other studies using pDNA vaccines in tattooing, we principally focused on the humoral response and not on the generation of specific CD8+ cells [28,29,30]. The elicited antibodies against PfRH5 showed a solid inhibitory effect on blood stage *P. falciparum* turning the tattooing of DNA complexed with cationic liposomes into a valid avenue for antibody production, considering also that PfRH5 is a difficult-to-produce antigen. In other studies that tested PfRH5-based vaccines, similar growth inhibitory effects were obtained. For example, Douglas and colleagues observed a 40% inhibition when using 625 µg/mL purified IgG raised by adenovirus-vectored PfRH5 immunization [36], which is in the same range as we observed with 300 µg/mL murine IgG. The antibody response may be further enhanced by the use of a prime-boost regimen of immunization, for example by immunizing with pDNA followed by recombinant adenovirus, as done successfully with other malarial antigens [37]. In addition, the genetic fusion of antigen-encoding DNA fragments to HBs possibly leads to the intracellular production and secretion of virus-like particles with a high surface density of antigens. Importantly, the first malaria vaccine in broader use in different sub-Saharan countries consists of the circumsporozoite repeat sequence fused to HBs [38]. Also, the *Plasmodium vivax* merozoite antigen PvMSP1 fused to HBs formed virus-like particles and also showed high immunogenicity [39]. A similar *Plasmodium chabaudi*-derived construct was partially protective when intramuscularly delivered as a naked DNA vaccine [40]. In addition, we envision that the results shown here using an encapsulated DNA vaccine together with skin tattooing immunization can be transposed to RNA vaccines [41,42], thus avoiding potentially harmful DNA integration. Finally, the use of micro-needle patches as a one-shot-vaccine may still enhance immunogenicity while avoiding the repeated use of needles.

## 5. Conclusions

The combination of the promising, hard-to-produce vaccine antigen PfRH5 with particle-forming HBs and delivery as a DNA vaccine by the minimally invasive intradermal tattooing method is an effective method to elicit functional antibodies against PfRH5 and may be applicable to other relevant antigens.

## Figures and Tables

**Figure 1 vaccines-08-00619-f001:**
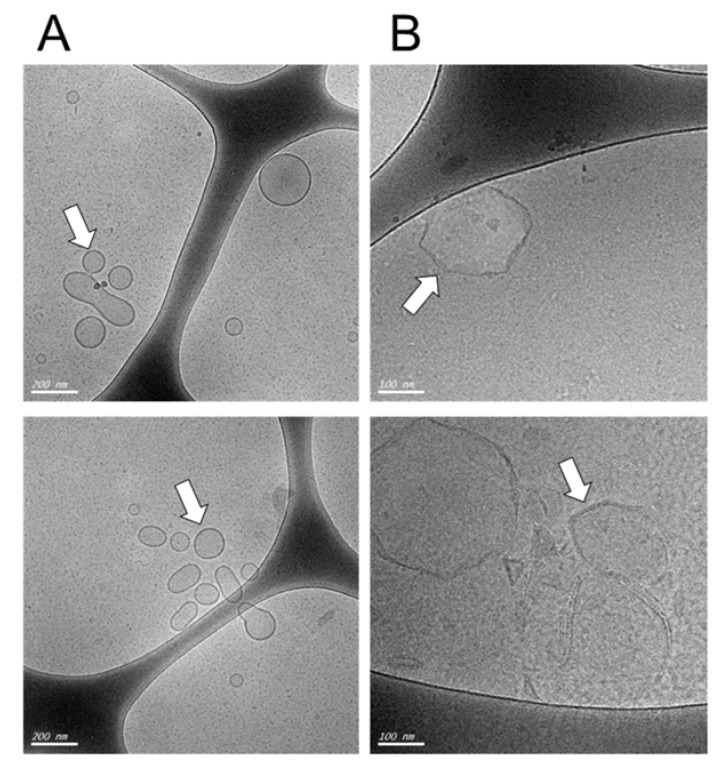
Visualization of cationic liposomes before (**A**) and after (**B**) plasmid loading. The cationic liposomes were visualized by Cryo-TEM microscopy at different scales, in (**A**), the scale bar is 200 µm, in (**B**) the scale bar is 100 nm. The arrows indicate the liposomal structures. Note that the liposomes show a change in their initial smooth surface to a more rugged appearance after plasmid loading.

**Figure 2 vaccines-08-00619-f002:**
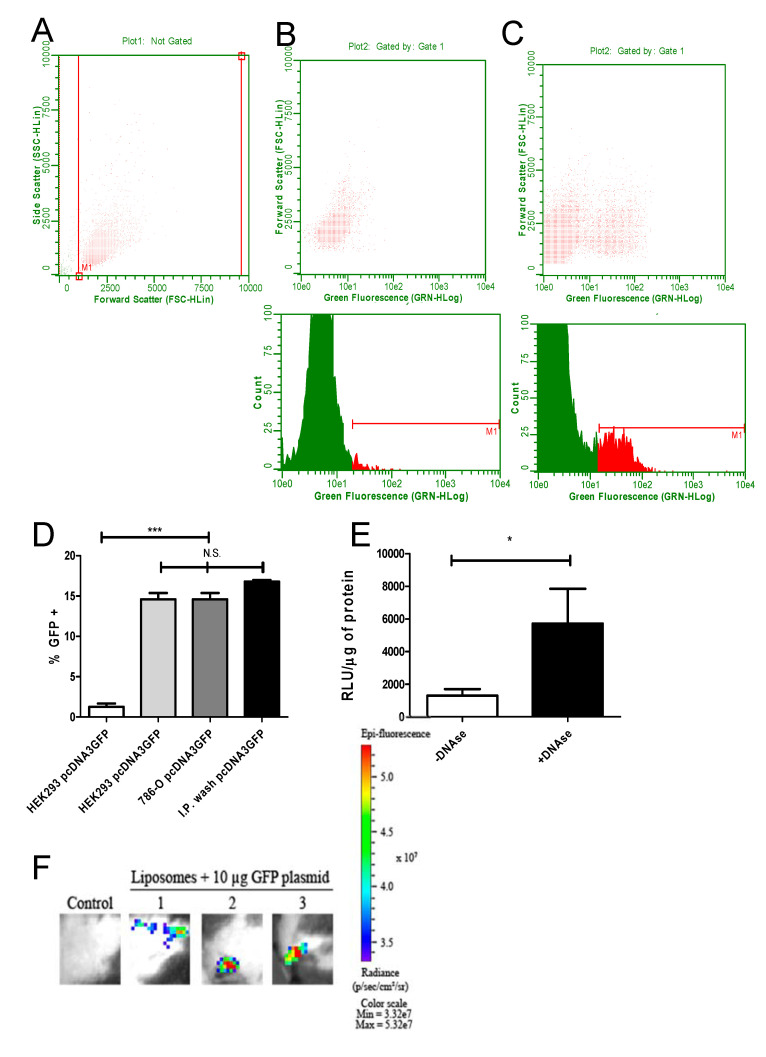
Efficient transfection of cells with green fluorescent protein (GFP) and luciferase-encoding plasmids encapsulated in cationic liposomes. To analyze the transfecting capacity of DNA-loaded cationic liposomes, HEK293T and 786-O cells were transfected in vitro as described in the Methods section. BALB/C mice were inoculated i.p. with encapsulated pcDNA3-GFP. Then, 24 h after transfection, GFP fluorescence was measured in trypsinized cells or peritoneal cells via cytometry. The gates used for size and side scattering analyses are shown in (**A**). In (**B**), cells exposed to empty liposomes (no GFP) are shown, in (**C**), cells transfected with encapsulated pcDNA3-GFP are depicted. The upper panel shows the dot plot while the lower panel depicts the histogram from the same sample analysis (HEK293 cells used in this experiment). In (**D**), the results of three independent experiments for each group are summarized, comparing the in vitro transfection from HEK293T and 786-O with the in vivo transfection (cells washed from the peritoneal region 24 h after transfection). HEK293T cells transfected either with encapsulated pcDNA3-GFP (second bar from the left), encapsulated pcDNA3-GFP transfected 786-O cells (third bar from the left) or washed, in vivo transfected peritoneal cells (bar on the right) showed similar fluorescence, significantly different from the HEK293T control (transfected with naked pcDNA3-GFP, left bar, significance was tested with ANOVA test, *** = *p* < 0.005). In (**E**), the influence of pcDNA3-luc DNA on the outside of cationic liposomes is shown after transfection, submitting cationic liposomes after loading to DNAse 1 treatment or not. Note that liposomes treated with DNAse showed a stronger transfection capacity, which is seen as a higher light emission mediated by luciferase expression. Relative light units were normalized as Relative Light Unit/µg. This experiment was done in three different samples in triplicate, analyzed by a paired T test with * = *p* < 0.05. (**F**) To validate the intradermal route for immunization, animals were tattooed with cationic liposomes loaded with pcDNA3-luc and control animals were tattooed with pcDNA3-PfRH5. The transfection efficiency was analyzed in an IVIS equipment 24 h after the tattooing process. The epi-fluorescence quantification of all (n = 3 per group) animals was measured as 5.24 × 10^8^ with a standard deviation of 1.5 × 10^8^.

**Figure 3 vaccines-08-00619-f003:**
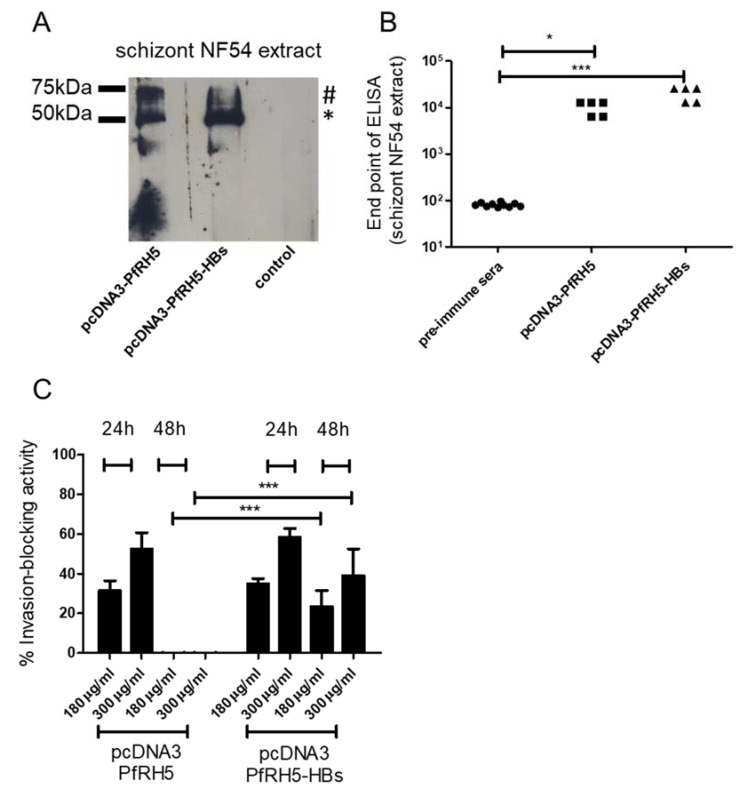
Antibody response after intradermal tattooing with plasmids encoding PfRH5 (*P. falciparum* reticulocyte binding protein homologue 5) or PfRH5 fused to small hepatitis B virus envelope antigen (HBs). (**A**) Sera from pcDNA3-PfRH5 and pcDNA3-PfRH5-HBs immunized mice were used to recognize native PfRH5 in extracts from *P. falciparum* NF54 schizonts. As a control, pre-immune sera from animals were used. Note that both the processed (*) and unprocessed (#) PfRH5 species were detected. Serum pools diluted 1:500 were employed in all experiments. In (**B**), endpoint ELISAs were done to measure antiPfRH5 IgG titers in pcDNA3-PfRH5 and pcDNA3-PfRH5-HBs-immunized mice. While pcDNA3-PfRH5-HBs showed values with a mean of 20480 s.d. +/− 7011, animals immunized with pcDNA3-PfRH5 showed values of 10240 s.d. +/− 3505. Assuming equal variance and non-parametric values, a statistically significant difference was observed between immunized-group sera and pre-immune sera (Kruskal–Wallis Test, * = 0.05, *** = 0.0004). In (**C**), the inhibitory activity of IgGs from antisera was monitored in in vitro growth assays. Data are shown as the percentage of growth inhibition of parasite proliferation (measured by flow cytometry) by purified IgG from immunized mice. A dose dependent assay with controls for each column was used. As controls, cultures grown in the presence of pre-immune sera pooled from each group at the maximum quantity were used (300 µg/mL). The inhibition was calculated as the decrease of parasitemia compared to the control sera-treated cultures, as described in the Methods section. The inhibitory effect of pcDNA3-PfRH5-HBs immunized mice sera after 48 h were significantly higher than that of pcDNA3-PfRH5 mice sera (two-way ANOVA with Bonferroni post-tests), while no statistically significant differences were found between all other groups. The values for 24 h and 48 h treatment are shown. As before, IgG from five animals per group were used.

**Table 1 vaccines-08-00619-t001:** Zeta potential (mV), polydispersity index and particle size (nm) of DDAB-Chol liposomes. These parameters were analyzed in triplicates of cationic liposomes alone or encapsulating 1 μg of plasmid DNA after loading.

Formulation	Liposomes (DDAB+DC Cholesterin)	8 nM of Lipid/µg of DNA Content
Zeta Potential (mV) Average ± SD	14.15 ± 1.22	−16.95 ± 1.55
Polydispersity index Average ± SD	0.225 ± 0.02	0.250 ± 0.01
Diameter (nm) Average ± SD	57.94± 0.85	110.54 ± 0.75

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
