# Peer review of "A DNA Vaccine Encoding Plasmodium falciparum PfRH5 in Cationic Liposomes for Dermal Tattooing Immunization"

_vaccines, 2020, doi:10.3390/vaccines8040619_

Round 1

Reviewer 1 Report

Authors did all the requested corrections and explanations. However, they still did not include positive control on Figure 3b (crucial figure/data in the paper). Many researchers are skipping such control and then claim that their vaccine is good but without proper comparison, it is difficult to claim it.

Author Response

In the first evaluation, a negative control was requested and we provided it in the revised Figure 3 B. These are antiSchizont/Merozoite extract IgG titers of the pre-immune sera. We understand that a positive control would require the immunization with a correctly folded recombinant PfRH5 protein produced in eucaryotic cells. We do not have this kind of protein to generate this very specific positive control. We don’t claim that our vaccine is better than other vaccines based on PfRH5, but we show in immunizations that its performance is comparable to other PfRH5-based vaccine approaches. In this sense, there is no reliable “gold standard” positive control for PfRH5, perhaps Oxford’s Adenovirus based PfRH5 vaccine (Douglas et al. 2011, new ref 36) which performs comparably. We now point to this in the discussion.

Reviewer 2 Report

All the previous comments have been addressed by the authors.

Author Response

We have gone through the manuscript again and made several changes which should have improved English mistakes.

Reviewer 3 Report

This manuscript describes the use of a strategy that combines using dermal tattooing with DNA encapsulated in cationic liposomes for improved vaccine potency, as determined by increased antibody production in mice in vivo studies. The authors also report an interesting finding that degradation of surface-adsorbed DNA improved transfection efficiency in vitro. It is generally well written and it is recommended for publication with the following minor revisions/questions:

  1. Figure 2E with DNAse treatment to remove adsorbed DNA shows interesting data. A statement regarding this result in the abstract would be beneficial.
  2. Also, were the particles that were injected in vivo treated with DNase prior, to provide /confirm increased transfection? If not, then it may be worth pursuing in the future. If it is, then is the increased Ab response using the tattooing method (and DNase treatment) vs previous IP injection solely due to injection method alone, or is it due to the change in particle surface chemistry?
  3. Ln 336: “Given that titers of antibodies were similar between studies, this route of delivery therefore appears as efficient as intraperitoneal immunization[14] with the difference that only three doses of tattoo-applied DNA-liposomes elicited the same antibody response as four ip immunizations.” In the previous study (Ref 14 by the same authors), what was the antibody response after 3 IP immunizations? This would be a more direct comparison to highlight the benefits of tattooing via DNA-cationic liposomes.
  4. It should be stated more clearly in the introduction and abstract that the novelty of this paper lies in the combination of using both cationic liposomes to deliver DNA and via tattooing – it was unclear from the text what exactly was done in previous reports (e.g either non-tattooing methods, or did not use encapsulated DNA in cationic liposomes).
  5. Lines 237 : The captions are unclear. “B, a typical transfection control with a plasmid encoding solely PfRH5 for in vivo analyses and plasmid alone without cationic liposomes for in vitro transfection, in C, cells transfected with pcDNA3-GFP”
  6. Fig 2D : X-axis labels are not aligned. Also, which represents the ‘naked’ pcDNA-GFP- transfected HEK 293 cells? Statistical analysis on the graph show that there is no significant difference between the naked and encapsulated pcDNA-GFP for HEK293T cells (2 bars on the left).

Author Response

  1. Figure 2E with DNAse treatment to remove adsorbed DNA shows interesting data. A statement regarding this result in the abstract would be beneficial.

This is an interesting aspect. See below.

  1. Also, were the particles that were injected in vivo treated with DNase prior, to provide /confirm increased transfection? If not, then it may be worth pursuing in the future. If it is, then is the increased Ab response using the tattooing method (and DNase treatment) vs previous IP injection solely due to injection method alone, or is it due to the change in particle surface chemistry?

Regarding mentioning the DNAse treatment in the abstract, we believe that the abstract should contain only the principal, vaccine-relevant results from the immunization tests. Since we have not tested DNAse treatment before immunizations we do not know if this modifies the in vivo transfection effect and therefore we chose to not modify the abstract. It may be expected that foreign DNA of bacterial origin (CpG containing) will trigger a Toll-like receptor-mediated innate response. On the other hand, it is expected that DNAses present in serum rapidly digest outside-exposed DNA. We now briefly mention this point in the discussion (lines 644 f).

  1. Ln 336: “Given that titers of antibodies were similar between studies, this route of delivery therefore appears as efficient as intraperitoneal immunization[14] with the difference that only three doses of tattoo-applied DNA-liposomes elicited the same antibody response as four ip immunizations.” In the previous study (Ref 14 by the same authors), what was the antibody response after 3 IP immunizations? This would be a more direct comparison to highlight the benefits of tattooing via DNA-cationic liposomes.

The antibody response after 3 ip injections in the previous study was 102-103 (unpublished data) and this was the reason why four immunizations were applied at that time. Also, in the previous study, an interferon-gamma encoding plasmid was co-injected in the fourth dose turning this direct comparison difficult. We now introduced a reference (novel ref 36) of a study dealing with a PfRH5 based vaccine and growth inhibition assays for comparison (lines 676 f).

 It should be stated more clearly in the introduction and abstract that the novelty of this paper lies in the combination of using both cationic liposomes to deliver DNA and via tattooing – it was unclear from the text what exactly was done in previous reports (e.g. either non-tattooing methods, or did not use encapsulated DNA in cationic liposomes).

This was modified accordingly (modified in abstract and introduction).

  1. Lines 237: The captions are unclear. “B, a typical transfection control with a plasmid encoding solely PfRH5 for in vivo analyses and plasmid alone without cationic liposomes for in vitro transfection, in C, cells transfected with pcDNA3-GFP”

This was corrected. We apologize for this lapse.

  1. Fig 2D: X-axis labels are not aligned. Also, which represents the ‘naked’ pcDNA-GFP- transfected HEK 293 cells? Statistical analysis on the graph show that there is no significant difference between the naked and encapsulated pcDNA-GFP for HEK293T cells (2 bars on the left).

We thoroughly corrected the legend which contained multiple misleading information. Unfortunately, GraphPad Prism does not allow to align legends to the bottom – legends are aligned along the X-axis.

This manuscript is a resubmission of an earlier submission. The following is a list of the peer review reports and author responses from that submission.

Round 1

Reviewer 1 Report

In the paper “A DNA Vaccine Encoding Plasmodium falciparum  PfRH5 in Cationic Liposomes for Dermal Tattooing  Immunization” authors described DNA vaccine incorporated into liposomal delivery system. Interestingly, they were delivery it using tattooing technique and produced relatively strong immune responses.

Major issues:

  1. Authors should discuss how their paper is different from Mol Ther Methods Clin Dev. 2017 Dec 15; 7: 1–10. doi: 1016/j.omtm.2017.08.004 DNA-Loaded Cationic Liposomes Efficiently Function as a Vaccine against Malarial Proteins

  1. Statistical analysis is wrong, simple T-test or one way ANOVA is not enough. Two way ANOVA or one way ANOVA plus posthoc test are required.
  2. Figure 3B. Crucial problems.
  • ELISA is done without any controls, it must include negative control, PBS immunized mice (to see antibody specificity, background, etc.), any positive control, e.g. injected PfRH5 protein-based vaccine (to see real efficacy of current system compare to some standard). Author mentioned intraperitoneal immunization and claimed that their titers are similar, it can be only compared when done by the same group in the same experiment.
  • ELISA must be done against protein used for immunization, not only against extract, to examine potential non-vaccine induced antibody bonding to extract.
  • Finally, presented differences on the Figure 3B will be not statistically significant when proper statistical method is used.
  • In the same manner (with appropriate controls) must be presented experiment shown on Figure 3C.

 Minor:

  1. Material and Methods point 2.2: Minor but important. Description how to produce liposomes is very poor. What is DC? , DDAB name is wrong; it is not clear how many milligram of each lipids is added to how many milliliters of solvent? Ratio of lipids is not giving this answer. DDAB is provide as concentration (nM) and DNA as quantity (microg), so it is not possible to repeat experiment. Everything should be reported as quantity in mmol only or micrograms only. Solvent quantity should be provided too.
  2. 7: why ELISA plates were coated with extract and not protein used for immunization?
  3. Where is Figure 1C (or 1D)?

Without proper controls (positive and negative), mentioned above, this manuscript could not be accepted.

Reviewer 2 Report

The authors developed nano-sized liposomes for delivering DNA plasmid-based vaccine for Plasmodium falciparum. For delivering they used dermal tattooing which is adding significant novelty to this work. While the study is well designed and the manuscript is well written, there are few minor concerns that need to be addressed:

  1. The authors mention that “The diameter of the liposomes was expected to be lower than 100 nm, as seen in Cryo-TEM”, however the Cryo-EM images reveal the size to be more than 100 nm. Kindly restate that. Also, DLS measure hydrodynamic diameter so is usually more than diameters observed in microscopic images, however here it is the contrary. An explanation on the difference in size between EM images and DLS is required.
  2. The rationale for selecting kidney cell lines for in vitro assay needs to be provided.
  3. Figure 3 C is confusing. Why is the % invasion activity increasing when going from 180 to 300 ug/ml and decreasing when going from 24 hr to 48 hr? A short discussion on this is required to clearly convey the message. Also statistical significance needs to be added in Fig 3C.